# Super-Multi-Junction Solar Cells—Device Configuration with the Potential for More Than 50% Annual Energy Conversion Efficiency (Non-Concentration)

**Kenji Araki [1,\*], Yasuyuki Ota [2] , Hiromu Saiki [3], Hiroki Tawa [3], Kensuke Nishioka [3] and Masafumi Yamaguchi [1]**

[1]  Toyota Technological Institute, Nagoya 468-8511, Japan; masafumi@toyota-ti.ac.jp
[2]  Organization for Promotion of Tenure Track, University of Miyazaki, Miyazaki 889-2192, Japan; y-ota@cc.miyazaki-u.ac.jp
[3]  Faculty of Engineering, University of Miyazaki, Miyazaki 889-2192, Japan; hk14017@student.miyazaki-u.ac.jp (H.S.); hk14028@student.miyazaki-u.ac.jp (H.T.); nishioka@cc.miyazaki-u.ac.jp (K.N.)
\*  Correspondence: cpvkenjiaraki@toyota-ti.ac.jp; Tel.: +81-52-809-1830



**Featured Application: This technology is expected to be applied to vehicle-integrated photovoltaic and therefore the installation area is limited, but high performance is demanded.**

**Abstract:** The highest-efficiency solar cell in the efficiency race does not always give the best annual energy yield in real world solar conditions because the spectrum is always changing. The study of radiative coupling of concentrator solar cells implies that efficiency could increase by recycling the radiative recombination generated by the surplus current in the upper junction. Such a configuration is called a super-multi-junction cell. We expand the model in the concentrator solar cell to a non-concentrating installation. It is shown that this super-multi-junction cell configuration is robust and can keep maximum potential efficiency (50% in realistic spectrum fluctuation) for up to 10 junctions. The super-multi-junction cell is also robust in the bandgap engineering of each junction. Therefore, a future multi-junction may not be required for tuning the bandgap to match the standard solar spectrum, as well as relying upon artificial technologies such as epitaxial lift-off (ELO), wafer-bonding, mechanical-stacking, and reverse-growth, but merely uses upright and lattice-matching growth technologies. We present two challenging techniques; one is the optical cap layer that may be the directional photon coupling layer in the application of the photonics technologies, and another is the high-quality epitaxial growth with almost 100% radiative efficiency.

**Keywords:** tandem; solar cell; multi-junction; performance ratio; spectrum; modeling; radiative coupling; luminescence coupling

## 1. Introduction

Solar panels with more than 40% power conversion efficiency in the real world will change our society, including running most electric vehicles on solar energy [1]. The potential for conversion efficiency of solar cells is one of the most popular research topics in photovoltaic science and has been studied intensively by many people, suggesting a bright future for the potential of photovoltaic energy conversion [2–4]. This is based on solid scientific background with ideal but trustworthy preconditions. However, the materials and processes in the real world are not ideal, and the record efficiency values

for photovoltaic cells are less than the values predicted by the scientists, for example, 28.5 % for Si solar cells [2,5,6]. For example, Yamaguchi et al. predicted more than 45% efficiency in field concentrator solar cells intensively studied for the application of CPV (concentrator photovoltaic) [2], but the highest efficiency ever achieved was 44.2% in 2013 by the Sharp Corporation [5,6]. More recently, research based on the practical limit of material improvement to various materials like Si, III–V, II–VI thin films, organic, and Perovskite, as well as various configurations such as quantum dots, hetero-junction, and multi-junction, has been published [7–11]. These kinds of efficiency-limit studies tend to present a decreasing record number following the improvement of the model, namely by increasing constraints and taking into account inherent limitations (which are small but non-negligible). However, taking the example of energy conversion efficiency, namely efficiency of conversion from sunlight (ASTM G173 AM1.5G standard solar spectrum) to electricity power, the highest-efficiency solar cells are a group of multi-junction cells [1,5–7].

The principles of multi-junction cells were suggested by Jackson in 1955 [12], and Wolf et al. investigated them in 1960 [13]. However, the efficiency of multi-junction cells did not make significant progress until 1975 because of inadequate thin-film fabrication technologies. The liquid-phase and vapor-phase epitaxy brought AlGaAs/GaAs multi-junction cells into the 1980s, with tunnel junctions by Hutchby et al. [14] and metal interconnections by Ludowise et al. [15], Flores [16] and Chung et al. [17]. Fan et al. predicted efficiency of close to 30% at that time [18], but this was not achieved because of difficulties with high-performance, stable tunnel junctions [19] as well as oxygen-related defects in the AlGaAs at that time [20]. Yamaguchi et al. developed high-performance, stable tunnel junctions with a double-hetero (DH) structure [21]. Olson et al. introduced InGaP for the top cell [22], and Bertness et al. achieved a 29.5% efficiency by a 0.25 $cm^2$ GaInP/GaAs multi-junction cell [23]. Recently, 37.9% efficiency and 38.8% efficiencies have been achieved with InGaP/GaAs/InGaAs 3-junction cell by Sharp [24] and with a 5-junction cell by Spectrolab [25].

Historically, high-efficiency multi-junction cells have been used for concentrator photovoltaic (CPV) cells. The energy conversion efficiency substantially increases under concentration operation [26]. Significant cost reduction was predicted in the 1960s [27]. The Wisconsin Solar Energy Center investigated the performance of solar cells under concentrated sunlight [27]. Research and development programs under the DOE (US Department of Energy), EC (European Commission), and NEDO (New Energy and Industrial Technology Development Organization, Japan) realized high conversion efficiencies using the CPV module and system. A 44.4% efficiency was demonstrated with the InGaP/GaAs/InGaAs 3-junction concentrator solar cell by Sharp [24]. The CPV system increased its installation in dry areas of the world after 2008. By 2017, total installation in the world reached 400 MW [28].

The outdoor performance of multi-junction solar cells for CPV application has been intensively analyzed, and the most significant loss is known as the spectrum-mismatching loss [28–37]. This is caused by the fact that the solar spectrum is not always the same as the designed one (typically, ASTM G173 AM1.5D spectrum for CPV application). The sub-cells in the multi-junction cells are electrically connected in series. The spectrum shift hampers the balance of the output current from the sub-cells, and the sub-cell with the smallest output current constrains the total output current according to Kirchhoff's law. This type of loss is called "spectrum-mismatching loss." Spectrum-mismatching loss is an inherent loss for all types of multi-junction or multi-junction solar cell, regardless of CPV or normal flat-plate application, except for more than 3 terminal configurations where the output of the sub-cells is individually connected to the load. Please note that in every type of installation there is a variation of the solar spectrum by sun height and fluctuation of scattering and absorption of the air by seasonal effect, but its influence can be minimized by the improvement of the solar cell design [38–43].

Research into the robustness to spectrum change has been made in the past 20 years, including a computer model named Syracuse by Imperial College London [44–46]. For CPV applications, it was understood that the chromatic aberration of the concentrator optics enhanced the spectrum-mismatching loss [44–53]. However, such loss, coupled with the concentrator optics, could be solved by the innovation

of optics, including homogenizers and the secondary optical element (SOE) [54,55]. The remaining problems of the spectrum-mismatching loss have been overcome by the adjustment of the absorption spectrum of each sub-cell, including overlapping the absorption spectrum and broadening the absorption band to the zone of massive fluctuation.

Recently, a new configuration enhancing the radiative coupling among the sub-cells has been found useful for solving this inherent loss of the multi-junction cells. The first study was presented by Browne in 2002 [56]. However, his model was too simplified and overlooked the most important factor, namely a variation of atmospheric parameters. Later, Chen developed a power-generation model considering the variation of atmospheric parameters and quantitatively anticipated that the radiation coupling would be adequate for suppressing the spectrum-mismatching loss [57–60]. This idea was further developed by a group of authors [61–64]. However, the work of these authors was limited to the application of CPV because of the simplicity of spectrum and performance modeling.

Radiative recombination was also identified to impact the performance of the multi-junction cell, even in operation under standard testing conditions (not a dynamically changing spectrum like the outdoor spectrum). Taking an example of the research on Fraunhofer ISE [65], and later, by use of a rear-side mirror for the use of the recycled photon by radiative recombination, high open-circuit voltage, and 28.8% efficiency under 18.2 W/cm$^2$ concentrated irradiance was obtained [66]. The measurement and identification of radiative coupling and photon recycling was done in several types of solar cells, including GaAs cells [67], strain-balanced quantum well cells [68], and even emerging solar cells such as Perovskite solar cells [69]. Radiative coupling also affects the measurement of multi-junction solar cells, and it is often called luminescence coupling [70–72].

Recently, multi-junction solar cells have been considered for use in non-concentrating applications, including car-roof photovoltaics (PV) [1,9,73–85]. It was considered that most electric vehicles might be able to run by solar energy using a solar cell mounted on the car roof [1]. The area of the car roof is limited. Moreover, solar cells cannot be laminated to an undevelopable curved surface of the car body. It is difficult to entirely cover the car-roof surface. Therefore, extremely high performance is required for such an application.

Unlike CPV applications where the cell is always normal to the sun by solar tracker, and only receives direct sunlight, the non-concentration application needs to use a diffused component of sunlight from sky, ground reflection, and skewed solar rays, with a combination of direct and diffused components as a function of the sun orientation relative to the solar panel orientation.

This article describes the model of the spectrum variation behavior, with a contrast against previous research first [86–91]. Then, the model is validated by outdoor measurement. Finally, the potentials of performance impacted by a seasonal change of the spectrum are examined to understand whether the super-multi-junction configuration should be robust or not.

Since the target of this work is to identify the limit of the performance of solar cells under a realistic assumption of the spectrum, the material discussed in this work is the ideal one, not realistic with the current technology. However, it is far from realistic to attempt to change and control the solar spectrum to the ASTM G173 AM1.5G standard solar spectrum throughout the day time, and we will be able to improve the material quality to approach an ideal one. Although the discussion of solar cell performance relies on ideal material, by contrast, a realistic spectrum condition is different from the majority of research papers, and it is worth reconsidering the limit of solar cells under real the solar spectrum that most scientists sometimes forget.

## 2. Model

In this section, we present a model of multi-junction solar cells and super-multi-junction solar cells affected by the fluctuation of the spectrum. Since the solar spectrum is not affected by sun height (airmass), but is affected by many other climate and atmospheric conditions, we need to model the performance of multi-junction solar cells by probability model, namely the Monte Carlo method.

Next, we discuss how multi-junction solar cells behave under variation of atmospheric parameters with complex interactions with other climate and the sun-related variations.

### 2.1. What Is the Super-Multi-Junction Solar Cell

Although multi-junction cells have high efficiency, their performance ratio affected by spectrum variation is typically less than that of single-junction solar cells. This is due to spectrum-mismatching loss influenced by the variation of sun height [42,92] and atmospheric parameters [92,93]. The power output of conventional multi-junction solar cells constrained by spectrum-mismatching loss can be predicted, and we need a solution to minimize the damage.

The super-multi-junction cell uses enhanced luminescence coupling [63]. Assuming the extreme and the best case that every junction in a solar cell can couple to each other in radiation energy by the radiative recombination, the excess carriers in one junction can be recycled and transfer to the bottle-necked junction [63]. Figure 1 indicates the configuration of the super-multi-junction cell [63]. Please note that the optical cap layer in the super-multi-junction solar cell is for confining recycled photons, specifically to reduce the angle of the escape cone from the solar cell. Using radiative coupling, we may carry the energy that was to be lost by the surplus current by spectrum mismatching [63]. However, an excessive number of junctions is sometimes harmful, and there is no advantage in having more than four junctions [61,94]. The efficiency started to drop after more than 6 junctions in concentrator solar cells [61]. Calculation in the past was done in combination with the worst cases, such as a combination of worst-case atmospheric conditions, and perfect junctions (full absorption, no leakage) [61,94]. There may be a chance of reasonable compromise. Therefore, we need to develop a new model that considers an individual variation of atmospheric conditions and spectrum.

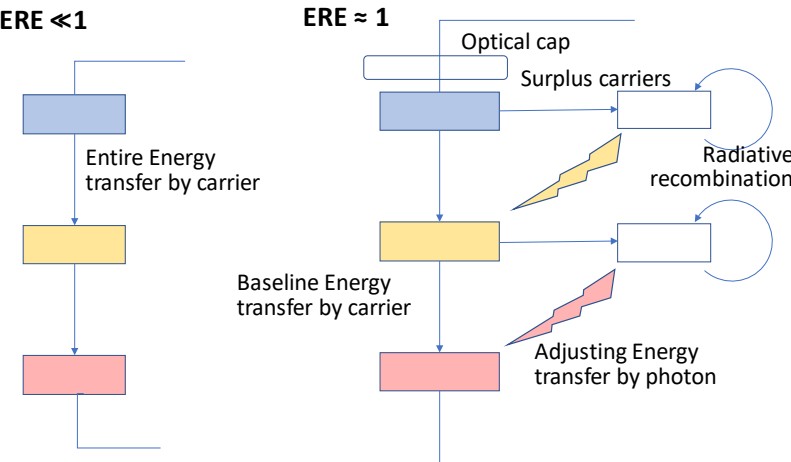

**Figure 1.** The energy flow of multi-junction cells. Left: Normal multi-junction cell; Right: Super-multi-junction cell. ERE—external radiative efficiency [61].

### 2.2. Monte Carlo Simulation for Analyzing the Annual Performance of Multi-Junction Cells

The design, performance analysis, and optimization calculation we used is a combination of the numerical optimization calculation and the Monte Carlo method (Figure 2) [63,92,94,95]. The vector of variables in numerical optimization is the set of bandgap energy at each junction. At each moment (time and date), the meteorological parameters and atmospheric parameters were given by random numbers (Monte Carlo method). The energy output across a year was summed and divided by the total irradiation, also calculated by the above meteorological and atmospheric conditions. The merit function for numerical optimization calculation is the annual average efficiency of the power conversion, directly coupled to the performance ratio. The initial value for numerical optimization calculation can be given by that of combination determined at the sun height on the winter solstice [96]. The optimized bandgap given by this method is identified to be close to the values given by the optimizing routine [96].

Considering that the target of this calculation is to identify the variation of output performance influenced by the different climate and spectrum in other years (Figure 2), the difference between the initial value and optimized value was not crucial; specifically, both had broad distributions [96], and difference between the initial value and optimized results were often invisible. Therefore, to save computation time, the first step of the flowchart in Figure 2 was optimized not by the annual dataset (365 days multiplied by the number of divisions of the time in the daytime) but by the representative sun height on the winter solstice.

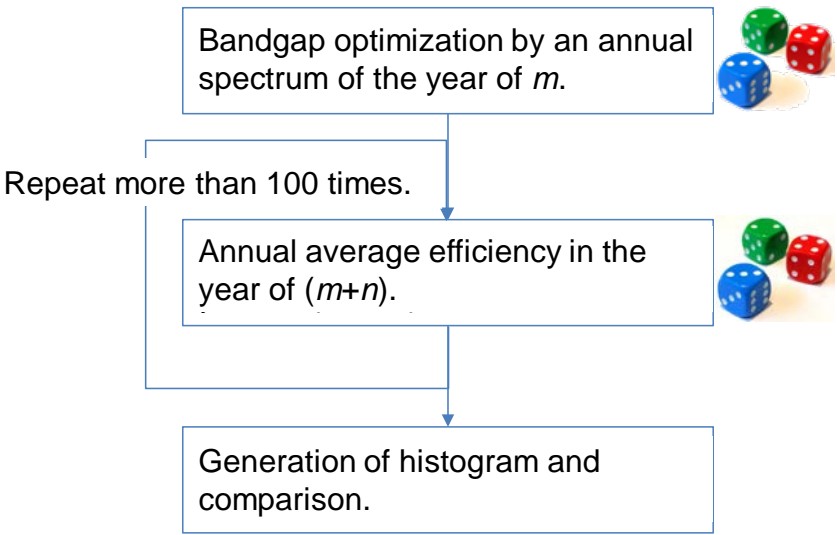

**Figure 2.** Flowchart of performance calculation using the Monte Carlo method.

With an increase in the number of junctions in the simulation in Figure 2, it may be the case that the efficiency of *i* number of junctions is higher than that of (*i* + 1) of the number of junctions. This case can be equivalently modeled by allowing the bandgap energy of the (*i* + 1)th junction to be equal or greater than that of the (*i*)th junction, but not allowing the bandgap energy of the (*i* + 1)th junction to be less than that of the (*i*)th junction.

### 2.3. Modeling Multi-Junction Solar Cells Affected by a Variety of Spectra

The dataset impacted by the fluctuation of the spectrum by a random number is given by either a histogram of the parameters [57–60] or superpositioning the random number provided by logarithmic normal distribution along the seasonal fluctuation trend lines of the atmospheric parameters [61,63,92,94,95]. The series resistance was assumed to be 1 $\Omega cm^2$, and the fill factor *FF* was calculated by the ratio of the spectrum mismatching—specifically, generating a correlation chart between calculated *FF* and the ratio of mismatching at first; then, general trend of these two parameters was fitted to the parabolic curve so that the *FF* is represented as the function of the spectrum-mismatching index. This step significantly accelerated the computation time. Otherwise, it would have been necessary to calculate every dataset of the output current and voltage (typically 100 points of the voltage and current of the *I-V* curve); then, the maximum power point would have been calculated by optimization problem. For the calculation of the performance ratio, this routine needed to be repeated for 12 representative days every month or 365 days (depending on the available solar irradiance data and computing time) multiplied by the number of division of the time in the daytime, or every 1 h, depending on the available solar irradiance database, for every attempt to seek the combination of the bandgaps of each junction in the optimization step. The external quantum efficiency was assumed to be unity by the wavelength corresponding to the bandgap of the junction. The angular characteristics in the photon absorption were assumed to be Lambertian. The open-circuit voltage at 1 $kW/m^2$ irradiance of each junction was assumed to the bandgap voltage minus 0.3 V, namely the best crystal quality

in the current epitaxial growth conditions [97]. Figures 3 and 4 summarize the assumptions in the calculation of the efficiency potential of the solar cell.

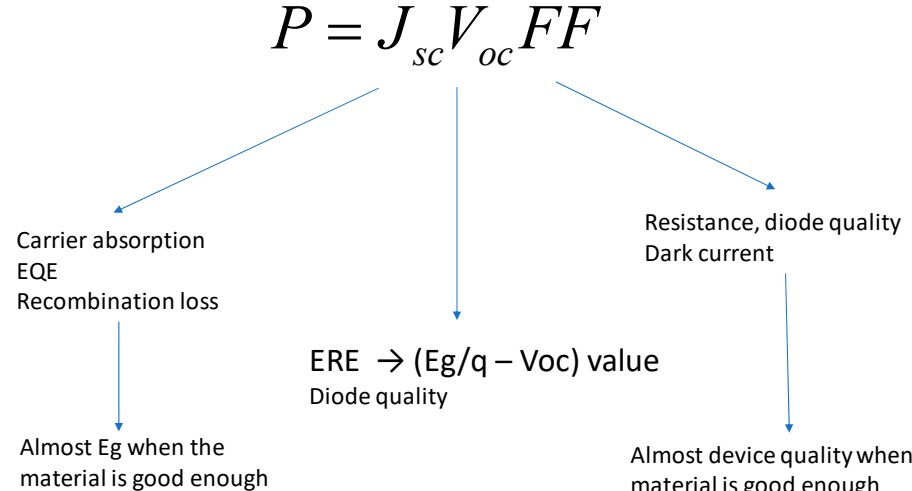

**Figure 3.** Diagram of how the output power of solar cells is calculated (composed of three factors).

### *Isc*:

Ideal absorption and rectangular EQE windows.
　100 % absorption bellow the bandend
　EQE: Unity bellow the bandend,
　EQE: Zero above the bandend.
Surface reflectance: Zero
Shading loss by electrodes: Zero
Angular response: Lambartian

### *Voc*:

Minus 0.3 V from the bandgap voltage at 1 kW/m² irradiance.
Junction temperature: always 27 C
Diode ideality factor: Unity (every junction)

### *FF*:

Parabolic approximation as a function of the spectrum mismatching ratio.
Series resistance: 1 Ωcm²
Leak resistance: Infinity
Diode ideality factor: Unity

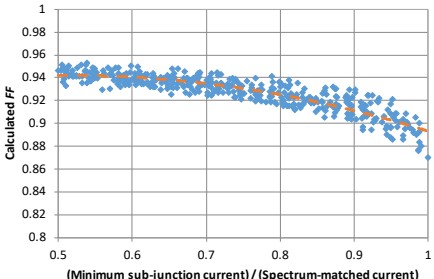

Example of 3J flat-plate given by Monte Carlo method

**Figure 4.** Assumptions in the calculation of the efficiency potential of the solar cell using three factors.

　　The analysis of concentrator solar cells was done in our previous research [61,63,92,94,95]. The calculation and analysis for concentrator solar cells was relatively simple because we did not have to consider angular effects combined with the mixture ratio of the direct and diffused spectrum of the sunlight. Moreover, concentrator solar cells generate power only under direct sunlight, but non-concentrating solar cells also generate power in diffused sunlight, so we have to model the solar spectrum in all kinds of climates. For extension to non-concentrating applications, we need to solve the complicated coupling of spectrum and angles (Table 1). The key parameters are atmospheric parameters that are dependent on each other. For example, different incident angle modifier and different orientation lead to a diverse mixture of direct and diffused sunlight. The atmospheric parameters were calculated by the spectrum, using a data-fitting calculation called the Spectrl2 model [98] at the University of Miyazaki [24,99]. The developed model for the analysis of non-concentrating solar cells is given in Figure 5 [99,100].

**Table 1.** The difference in performance modeling between concentrator PV and standard installation.

|  | CPV [1] | Normal Installation |
|---|---|---|
| Solar spectrum | Only direct | A mixture of direct, diffused from the sky, and reflection |
| Angle | Always normal | Varies by time and seasons |
| Spectrum by angle | Constant (only normal) | Needs consider coupling to angle |

[1] It only generates power only by direct solar irradiance using a 2-axis solar tracker.

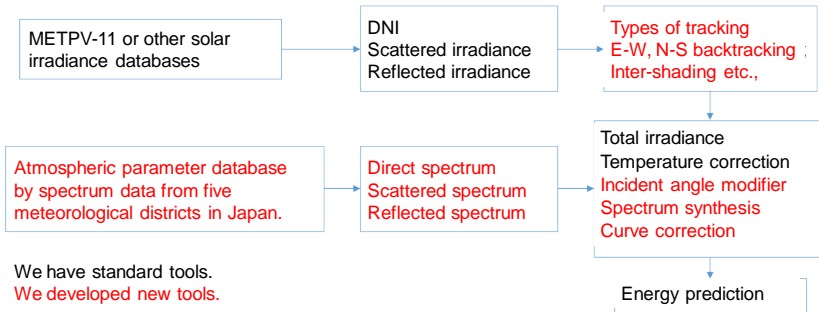

**Figure 5.** Modeling performance of the non-concentrating multi-junction solar cells considering the complicated spectrum and angle interaction described in Table 1. In this study, we only considered the flat plate, so that correction to the curved surface in the integrated tool was not applied.

## 3. Results

For analysis and optimization, thus anticipating the upper limit of the annual performance of both multi-junction solar cells and super-multi-junction solar cells under a non-concentration operation, we needed to verify the non-concentration operation model of the multi-junction solar cells affected by the spectrum (Figure 5). Therefore, we integrated the operation model (Figure 5) to bandgap optimization and distribution of the annual performance prediction using the Monte Carlo method (Figure 2). The integrated calculation was applied to normal multi-junction solar cells and super-multi-junction solar cells (Figure 1).

### 3.1. Validation of the Outdoor Operating Model for Non-Concentrating Multi-Junction Solar Cells

The calculated energy generation trend was compared to the PV module prototype using three-junction tandem cell monitoring at the University of Miyazaki. The validation of the model (Table 1 and Figure 3) was carried out with the cooperation of the University of Miyazaki [92]. The detailed structure of the module and outdoor performance can be found in the publication of Ota [101,102]. The solar cell used in the module was InGaP (1.88 eV)/GaAs (1.43 eV)/InGaAs (0.98 eV) inverted triple-junction solar cell. The InGaP top and the GaAs middle cell layers were grown on a GaAs substrate at first using metal organic chemical vapor deposition (MOCVD) technology, and then the InGaAs bottom cell (larger lattice-constant than GaAs) was grown. Deterioration of the crystal quality of the InGaP/GaAs layers was avoided before the growth of the buffer layer. After the growth of cell layers in an inverted order, cell layers were mounted on a handling substrate, and the GaAs substrate was removed. The module was assembled using these mounted cells, and its efficiency reached 31.17% under the standard testing condition [101,102].

The general trend between the model and measurement is shown in Figure 6. Although the model trend was generated by the values of average years from the meteorological and solar irradiance database (METPV-11), the seasonal pattern matched the measured performance very well. Please note that the measured trend of the non-concentrating operation of the high-efficiency three-junction solar cell (31.17% efficiency) demonstrates a strange fluctuation of performance that could not be explained by the conventional model, as shown in the right chart in Figure 6; however, the calculated trend by the new model (Table 1 and Figure 5) successfully explains the strange behavior affected by spectrum change coupled with angular characteristics.

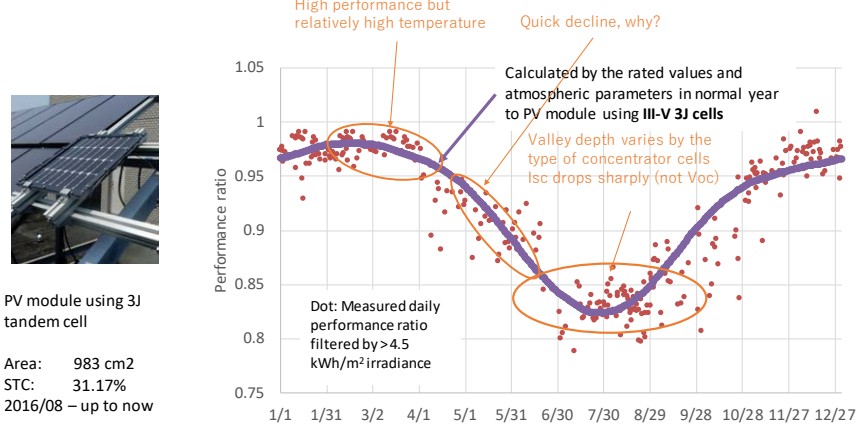

**Figure 6.** Comparison between the measured and modeled seasonal trends of the performance of the PV module using multi-junction solar cells [96]. Performance ratio can be calculated by the formula defined as PR = Yf/Yr, where PR is performance ratio, and Yf is the integrated energy yield of one day, and Yr is the nominal energy yield of one day calculated by the standard testing condition (STC) module efficiency and total insolation.

In the validation of this model, the critical parameter related to the calculation in the super-multi-junction solar cell is the degree of luminescence coupling between the middle junction and the bottom junction. Note the degree of radiative coupling from the middle cell to the bottom cell (typically 15%) is the key to the validation of the model, and we must consider its coupling; otherwise, the model (Figure 2) could not meet to the outdoor validation (Figure 7). The level of the coupling ratio of the middle junction (GaAs) was measured by Derkacs et al. as the function of the current level using a GaAs/GaInNAsSb two-junction cell, and the one corresponding to the non-concentration operation (14 mA/cm$^2$) was about 15% [103].

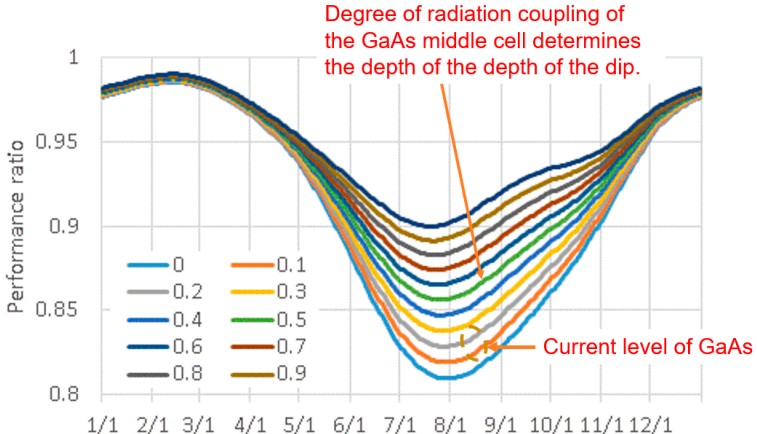

**Figure 7.** Recovery of the spectrum-mismatching loss due to water absorption in summer by enhancing the ratio of luminescence coupling between the middle junction and the bottom junction, added and modified from the original chart in [97]. The multiple colored lines correspond to the level of the luminescence coupling between the middle junction and the bottom junction, from the bottom to the top, 0%, 10%, 20%, . . . , 90%. Please note that the variation of the performance ratio impacted by the spectrum change was reduced by the increase of the level of luminescence coupling, but the right depth in summer corresponds to the ones of 10% and 20% of the luminescence coupling. Performance ratio can be calculated by the formula defined as PR = Yf/Yr, where PR is performance ratio, and Yf is the integrated energy yield of one day, and Yr is the nominal energy yield of one day calculated by the STC module efficiency and total insolation.

### 3.2. Normal Multi-Junction vs. Super-Multi-Junction; Practical Conditions

The design of the super-multi-junction cells by the worst-case atmospheric conditions can be done assuming both aerosol density and water precipitation.

The achievement in Section 3.1 implies that we can apply the model to the practical conditions by validated energy generation model of the multi-junction solar cell affected by the spectrum variation, considering complexed conditions listed in Table 1 and using the calculation flow in Figure 3. However, we need local data for both climate (solar irradiance) and atmospheric parameters. The model depends on local conditions and is not applied globally.

Another crucial point is that the distribution of atmospheric parameters, especially aerosol density, was worst for the general performance on multi-junction solar cells with more than three junctions, even though the airmass level (20° of latitude) is low. The worst-case distribution of the aerosol density is close to North India [57–60], and this region is known as one of the worst areas for energy generation in multi-junction solar cells in the field experience [104,105]. This is another reason that we need to develop an annual performance model based on realistic atmospheric conditions with a probability the realistic variations.

### 3.2.1. Modeling the Practical Spectrum Variation

To develop the operation model of the multi-junction solar cells affected by the probability distribution of the crucial parameters for the basic calculation flow in Figure 2, we defined the parameters by random numbers. Table 2 is the independent parent variables and Table 3 is the dependent variables calculated by the parent independent probability variables, considering local conditions.

**Table 2.** List of the probability parameters for modeling variation of annual performance (independent parent parameters).

| | Range and Type | Description |
|---|---|---|
| Variation factor in aerosol density | Normal distribution centered on 0 | Calculated by the residual errors in the measured point form the smooth trend line. |
| Variation factor in water precipitation | Normal distribution centered on 0 | Calculated by the residual errors in the measured point form the smooth trend line. |
| Variation factor in solar irradiance [1] | Ranged uniform distribution in [–1,1] | −1: Lowest irradiance year, 0: Normal year, 1: Highest irradiance year. The irradiance data is calculated by the linear coupling of three parameters depends on the value of the probability factor. The base irradiance data was given in 24 h × 365 days by METPV-11 and METPV-Asia database. |

[1] The same factor is applied both to direct and diffused sunlight.

**Table 3.** List of the probability parameters for modeling variation of annual performance (dependent parameters).

| | Parent Parameters | Description |
|---|---|---|
| Aerosol density | Variation factor in aerosol density | The variation factor gives a relative displacement from the trend line of the aerosol density. |
| Water precipitation | Variation factor in water precipitation | The variation factor gives a relative displacement from the trend line of water precipitation. |
| Direct irradiance | Variation factor in solar irradiance | Calculated by linear coupling of the data of the highest year, normal year, and the lowest year depends on the value of the probability factor. |
| Diffused irradiance from the sky | Variation factor in solar irradiance | Calculated by linear coupling of the data of the highest year, normal year, and the lowest year depends on the value of the probability factor. |
| The slope angle of the installation [1] | Both direct and diffused solar irradiance | Calculated by the optimization calculation given by the datasets of the solar irradiance affected by the variation factor in solar irradiance (parent parameter). |

[1] Meaning that the slope angle is determined simultaneously by the combination of the optimized bandgaps in the junctions by the measured one year irradiance (affected in the measurement in the first step in Figure 2).

The crucial probability parameters are the first two in Table 2. The distribution of these parameters was analyzed by the comparison between measured atmospheric parameters from the seasonal trend

lines. The seasonal trend lines of the atmospheric parameters, specifically aerosol density and water precipitation, are plotted in Figure 8. These were calculated by the data-fitting of the periodically observed solar spectrum line in a horizontal plane at the University of Miyazaki, Japan (N31.83°, E131.42°) [61,92,93,99,100,106]. Generally, the aerosol density is high in winter but low in summer, and the water precipitation, on the other hand, is high in summer. This trend can be seen for the entire region of Japan. However, there may be some regional characteristics. In Miyazaki, for example, a distinct peak in aerosol density appears in April that corresponds to the pollen of cedar and cypress trees.

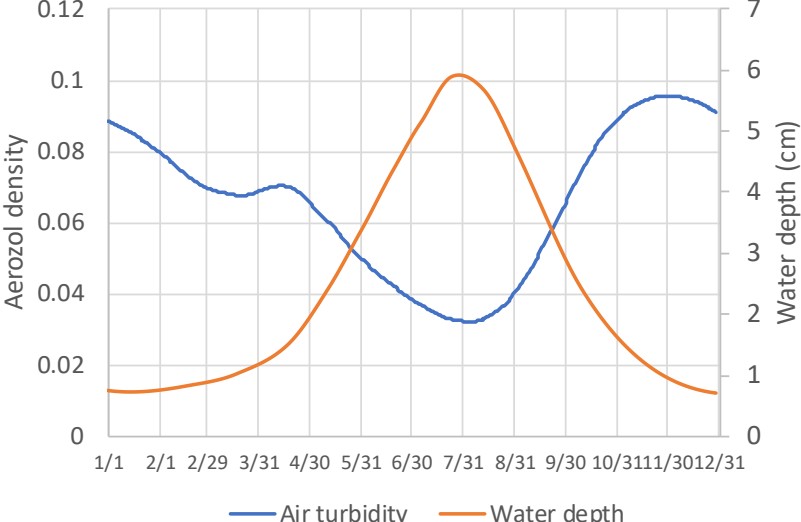

**Figure 8.** Seasonal fluctuation of the atmospheric parameters around the University of Miyazaki, taken by the curve-fitting method to the spectral profile modeled by Spectrl2 [95]. The trend line was defined by the local least-square-error method.

The fluctuation of the parameters from the trend lines can be modeled by the approximation of the distribution function of the residual error. The residual errors of the measured atmospheric parameters from the trend line (relative to the values in the trend line) are plotted in Figure 9.

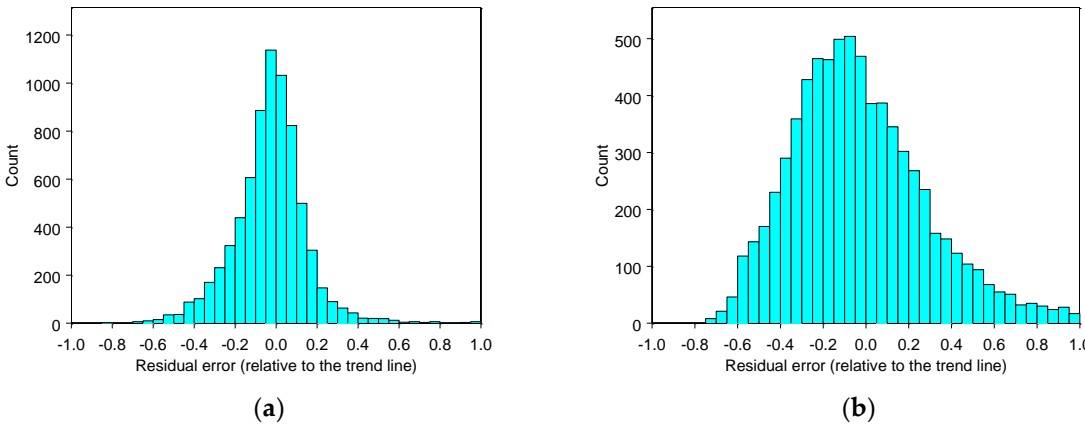

**Figure 9.** Histogram of the residual errors of the measured atmospheric parameters from the trend line (relative to the values in the trend line): (**a**) Aerosol density; (**b**) Water precipitation.

To find the best representative distribution, we used a Q–Q plot, i.e., a quantile–quantile plot that examines the values of two distributions (Figure 10). The best results were found in the normal distribution in both cases. In this plot, the *x*-axis corresponds to the values distributed to the normal distribution, and the *y*-axis corresponds to the measured values. If these two distributions are entirely

matched, the plotline will be along the 45° (y = x) line. The parameter sets of the normal distribution of the aerosol density and water precipitation were (0, 0.30) and (0, 0.38). The first term inside the parentheses is the mean value, and that of the second value is the standard deviation. We also examined the statistical adequateness by the one-sample Kolmogorov–Smirnov test [106]. The alternative hypothesis was "True: cumulative distribution function is not the normal distribution with given parameters, for example (0, 0.30) for aerosol density, with estimated parameters". The p-value in both cases was zero, implying that it is next to impossible to deny that both distributions of the relative residual errors of atmospheric parameters from the reference trend lines are different from the normal distribution. Therefore, we defined the probability parameters in the first two parameters in Table 1 (variation factor in aerosol density and variation factor in water precipitation) as the random numbers distributed normal distribution centered in zero and 0.30 and 0.38 standard deviations.

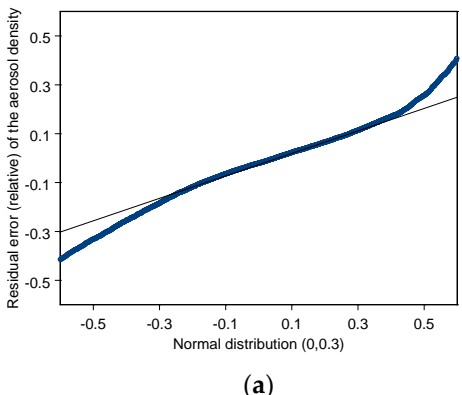
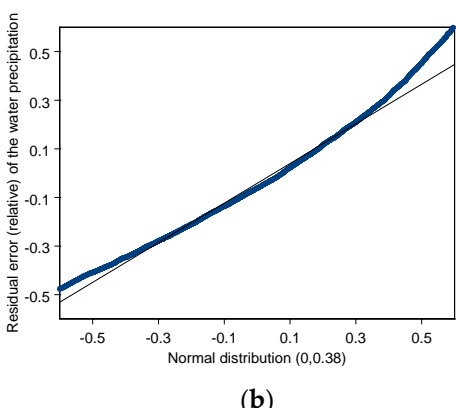

(**a**)                                            (**b**)

**Figure 10.** Quantile–quantile plot that examines the values of two distributions: (**a**) Aerosol density; (**b**) Water precipitation.

### 3.2.2. Computation Results of the Monte Carlo Simulation in the Practical Conditions

The distribution of the annual average efficiency of both a multi-junction solar cell and a super- multi-junction solar cell optimized by the spectrum in one year in Miyazaki is shown in Figure 11. The trend of the average of the annual average efficiency in each event in Figure 2, as well as the standard deviation of the distribution, is shown in Figure 12, overviewing the general efficiency trend after optimization. Please note that the spectrum for optimization was not the artificial standard spectrum (AM1.5G), but an accidental annual spectrum given by Monte Carlo simulation calculated by the flowchart in Figure 5, considering both seasonal and accidental fluctuation in the atmospheric parameters and fluctuation of the solar irradiance within the range of the highest and lowest irradiance in Miyazaki taken from the solar irradiance database of METPV-11. The underlying probability model for the calculation of the distribution of the average annual efficiency was given by the flowchart in Figure 2.

The normal multi-junction solar cell showed the broader distribution of the average annual efficiency depending on the spectrum in that year, as the increase of junction number. This is because the width of the absorbing spectrum band of each junction becomes narrower. This implies that the impact on the annual average efficiency by the spectrum-mismatching loss increases with the increase of the number of junctions. As a result, the annual average efficiency peaked at four junctions and then decreased with the increase of the number of junctions.

The super-multi-junction solar cell, on the contrary, showed narrower distribution, but it still shows a slightly broader distribution as the junction number increases. The annual average efficiency in the super-multi-junction solar cells is expected to reach 50% by 6–8 junctions.

An example of the distribution of the optimized bandgap energy of 10-junction solar cells is shown in Figure 13. The optimized bandgap was calculated according to the spectrum and other climate conditions given by random numbers, according to Figure 2. The histogram of the calculated

optimized bandgap energy in each junction is normalized so that the integral of the range becomes unity. The overlap of each peak does not mean that the higher bandgap junction has lower bandgap energy than that of the lower peak. It is constrained that the bandgap structure was equivalently modeled by allowing the bandgap energy of the ($i$ + 1)th junction to be equal or greater than that of the ($i$)th junction, but not allowing the bandgap energy of the ($i$ + 1)th junction to be less than that of the ($i$)th junction.

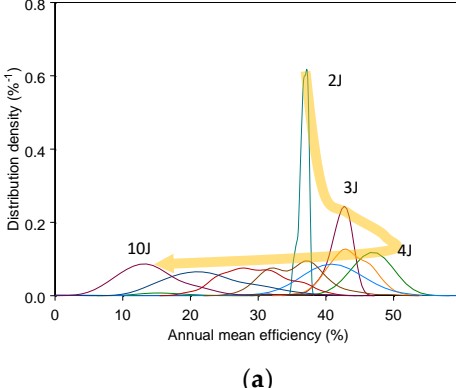 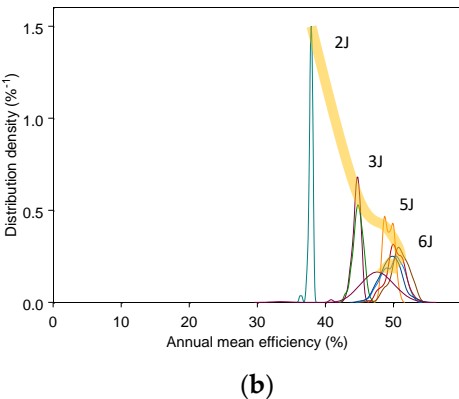

(**a**)        (**b**)

**Figure 11.** Optimization design result of the normal multi-junction solar cells (distribution of the annual average efficiency) under the worst-case combination of climate, atmospheric conditions, latitude, and orientation angle. The *y*-axis is normalized so that the integration of the distribution becomes unity: (**a**) Normal multi-junction solar cell; (**b**) Super-multi-junction solar cell.

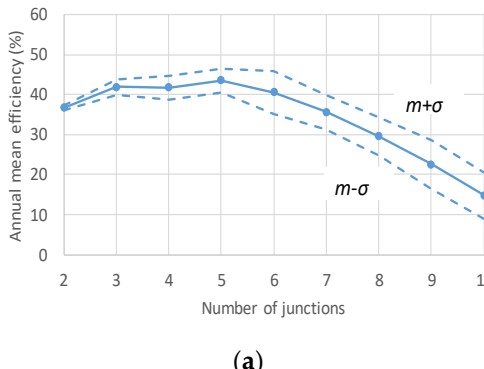 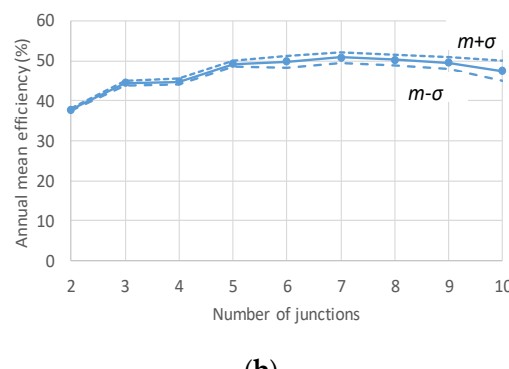

(**a**)        (**b**)

**Figure 12.** Optimization design result of the normal multi-junction solar cells (trend of an average of the annual average efficiency by variation of the spectrum) under worst-case combination of climate, atmospheric conditions, latitude, and orientation angle. *m* indicates average of the annual average efficiency, and *σ* indicates its standard deviation: (**a**) Normal multi-junction solar cell; (**b**) Super-multi-junction solar cell.

The most distinct difference of the super-multi-junction solar cell from the normal multi-junction solar cell is the level of the top junction. The distribution of the optimized bandgap energy of the top junction was substantially lower than that of the normal multi-junction solar cell. This is because the short-wavelength region of the sunlight is changeable by fluctuation of the aerosol scattering and the lower bandgap energy in the top junction is favorable in generating surplus current, so it compensates for the spectrum-mismatching loss by transferring the photon energy generated by the recombination using the surplus current of the top junction.

The set of the bandgap energy of the super-multi-junction solar cell is listed in Table 4. Unlike the current technology, the designed bandgap of each junction has a range, demonstrating that the super-multi-junction solar cell is robust to bandgaps.

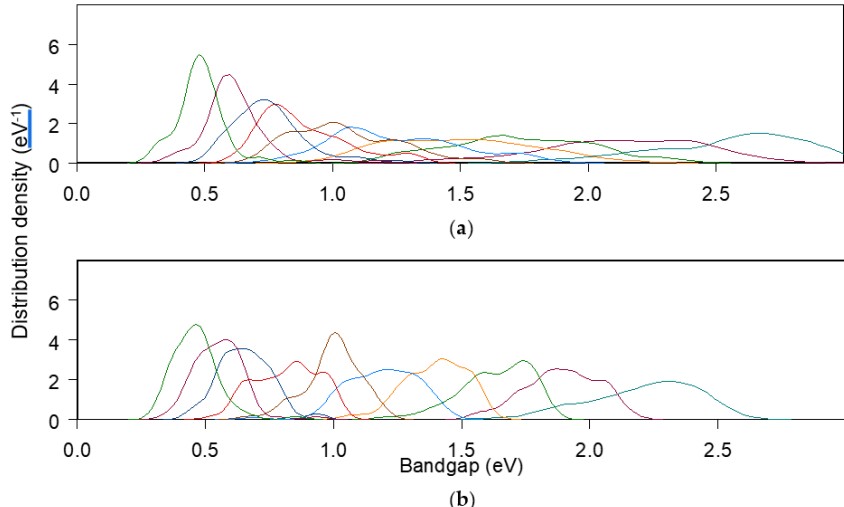

**Figure 13.** Distribution of the bandgap energy of the optimized (to the spectrum and other climate conditions given by random numbers according to Figure 2) multi-junction solar cells under the modeled fluctuation in the climate in Miyazaki, Japan (N 31.83°, E 131.42°). This is an example of 10 junctions. Please note that the histogram of the calculated optimized bandgap energy in each junction is normalized so that the integral of the range becomes unity. Also, note that the overlap of each peak does not mean that the higher bandgap junction has lower bandgap energy than that of the lower peak. It is constrained that the bandgap structure was equivalently modeled by allowing the bandgap energy of the $(i + 1)$th junction to be equal or greater than that of the $(i)$th junction, but not allowing the bandgap energy of the $(i + 1)$th junction to be less than that of the $(i)$th junction. The *y*-axis is normalized so that the integration of the distribution becomes unity: (**a**) Normal multi-junction solar cell; (**b**) Super-multi-junction solar cell.

**Table 4.** List of the set of the bandgap of the super-multi-junction solar cell.

| | Bandgap Energy (eV) from Top to Bottom Junction | | | | | | | | | |
|---|---|---|---|---|---|---|---|---|---|---|
| **2J** | 1.72 ± 0.03 | 1.12 ± 0.02 | | | | | | | | |
| **3J** | 1.89 ± 0.05 | 1.33 ± 0.07 | 0.89 ± 0.08 | | | | | | | |
| **4J** | 1.99 ± 0.07 | 1.47 ± 0.07 | 1.07 ± 0.09 | 0.73 ± 0.11 | | | | | | |
| **5J** | 2.11 ± 0.09 | 1.63 ± 0.07 | 1.27 ± 0.09 | 0.97 ± 0.08 | 0.72 ± 0.10 | | | | | |
| **6J** | 2.08 ± 0.15 | 1.68 ± 0.11 | 1.34 ± 0.11 | 1.07 ± 0.11 | 0.84 ± 0.11 | 0.66 ± 0.11 | | | | |
| **7J** | 2.17 ± 0.16 | 1.80 ± 0.11 | 1.48 ± 0.10 | 1.21 ± 0.12 | 0.99 ± 0.11 | 0.77 ± 0.12 | 0.62 ± 0.12 | | | |
| **8J** | 2.19 ± 0.16 | 1.84 ± 0.09 | 1.53 ± 0.11 | 1.28 ± 0.10 | 1.05 ± 0.10 | 0.86 ± 0.09 | 0.67 ± 0.09 | 0.55 ± 0.09 | | |
| **9J** | 2.25 ± 0.19 | 1.88 ± 0.13 | 1.61 ± 0.12 | 1.37 ± 0.11 | 1.13 ± 0.10 | 0.95 ± 0.10 | 0.70 ± 0.10 | 0.62 ± 0.08 | 0.52 ± 0.08 | |
| **10J** | 2.21 ± 0.21 | 1.89 ± 0.14 | 1.63 ± 0.14 | 1.40 ± 0.12 | 1.19 ± 0.14 | 1.00 ± 0.11 | 0.82 ± 0.12 | 0.66 ± 0.10 | 0.55 ± 0.09 | 0.46 ± 0.09 |

## 4. Discussion

In previous work, we have shown that super-multi-junction solar cells could solve low annual performance of concentrator photovoltaic systems affected by mismatching loss due to solar spectrum variation. Spectrum influence equally affects non-concentrating solar cells. However, the impact of the spectrum variation for non-concentrating applications needs to consider complex phenomena of direct, scattered, and reflected spectrum combined with angular effect. It was not appropriate to expand the model to non-concentrating applications.

We then tried to develop annual modeling performance of multi-junction solar cells, considering the spectrum (climate pattern, atmospheric parameters, sun-angle, airmass). The spectrum-enhanced performance model of multi-junction solar cells successfully explained the strange annual performance.

Then, we combined this model with previous work on optimization of the bandgap energy using the Monte Carlo method. The previous works on optimization and sensitivity to spectrum change relied on the distribution of atmospheric parameters, especially those of the worst case. This method

was too simple to describe the real fluctuation of the spectrum. For example, aerosol density and water precipitation had a distinct seasonal change that correlates with sun height and climate trends. The new probability model was developed by investigating the residual error distribution of atmospheric parameters that were identified to distribute the normal distribution.

The non-concentrating super-multi-junction solar cell was found to be robust and can keep almost the same maximum potential efficiency (50%) under the realistic conditions represented by Miyazaki, Japan (N 31.83°, E 131.42°).

Super-multi-junction solar cells are also robust to bandgap engineering of each junction. Therefore, future multi-junction cells may not be needed to tune the bandgap for matching the standard solar spectrum, or for relying upon artificial technologies like epitaxial lift-off (ELO), wafer-bonding, mechanical-stacking, and reverse-growth, but merely to be used for upright and lattice-matching growth technologies. However, we have two challenging techniques: one is the optical cap layer, which may be the directional photon coupling layer in the application of the photonics technologies, and the other is the high-quality epitaxial growth, with almost 100% of radiative efficiency (Figure 14).

**Figure 14.** Possibility of the future high-efficiency solar cell technology based on the implication from the super-multi-junction solar cell.

In comparison to the current level of the ERE of various solar cells that were collected by several authors [8,107–109], the requirement of the super-multi-junction solar cells is extremely high. The best-measured ERE, to the best knowledge of authors, is 35% [104]. This is far less than 100%. For the improvement of ERE, a typical and straightforward approach is to reduce threading dislocation density [110]. The target of the threading dislocation density is at least $10^3$ cm$^{-2}$, but should be as small as possible [110].

The function of the optical cap as the second technological challenge is the confinement of the photon. Any technological improvement in photon confinement typically using thin-film solar cells will be useful. A perfect solution is the use of the directional coupling of photons, typically used for communication technologies [111–114]. Although these optical devices are used in a narrow band of the wavelength, we expect that we may find useful hints from such different technological fields.

## 5. Conclusions

i.　　　Multi-junction cells: highest efficiency but lower energy yield.
ii.　　Super-multi-junction cell: compensation of spectrum-mismatching loss by sharing photons generated by radiation recombination due to surplus current of spectrum mismatching.
iii.　Annual performance: the model considering spectrum mismatching was validated and applied to super-multi-junction design.

iv.   Super-multi-junction solar cell performance: robust to spectrum change. Its annual average efficiency levels off at 50% with realistic spectrum fluctuation.

v.    Future multi-junction solar cells: may not be needed to tune the bandgap for matching the standard solar spectrum, or for relying upon artificial technologies such as ELO, wafer-bonding, mechanical-stacking, and reverse-growth, but merely to be used for upright and lattice-matching growth technologies.

**Author Contributions:** Conceptualization, K.A.; methodology, K.A.; software, K.A., H.S., and H.T.; validation, K.A., H.S., H.T., and Y.O.; investigation, Y.O. and M.Y.; data curation, H.S., H.T., and Y.O.; writing—original draft preparation, K.A.; writing—review and editing, K.A., and Y.O.; visualization, K.A., and Y.O.; supervision, Y.O.; project administration, K.N. and M.Y.; funding acquisition, K.N. and M.Y.

**Funding:** This research was funded by the New Energy and Industrial Technology Development Organization (NEDO) under the Ministry of Economy, Trade, and Industry (METI), Japan.

**Acknowledgments:** NEDO in Japan has partially supported this work.

**Conflicts of Interest:** The authors declare that there is no conflict of interest.

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
