# Peer review of "Super-Multi-Junction Solar Cells—Device Configuration with the Potential for More Than 50% Annual Energy Conversion Efficiency (Non-Concentration)"

_applsci, doi:10.3390/app9214598_

Round 1

Reviewer 1 Report

It seems that section 3.2 is missed in the actual version of the article, also this section is mentioned in lines 259, 265, 266. Without this section (description of the model) it is not possible to estimate the real value of the article.

Some important technical details concerning the multi-junction solar cells are missed. As an example, three -junction tandem cell from University of Miyszaki is mentioned very generally, without any description of the structure.

Very general consideration of the radiative coupling does not provide any insight on this problem from the material related point of view: it is not clear which materials/structures can provide most effective degree of the radiative coupling. Some concrete experimental facts, apart general statements like (typically 15%, line 245) should be provided.

Author Response

Dear Reviewer 1,

I appreciate your constructive suggestions. Due to the constrains of the browser, images were not pasted in this documents. I am wondering if you could find an attached file for my complete reply. 

It seems that section 3.2 is missed in the actual version of the article, also this section is mentioned in lines 259, 265, 266. Without this section (description of the model) it is not possible to estimate the real value of the article.

I appreciate that you have found a fatal flaw in the 1st manuscript. As you imagine, we made 3.2 at first, but we removed due to redundant contents. The float contexts in the line 259, 265, and 266 were the ones we forgot to correct. Now, the issue of the orphaned texts was solved.

 Some important technical details concerning the multi-junction solar cells are missed. As an example, three -junction tandem cell from University of Miyszaki is mentioned very generally, without any description of the structure.

I appreciate your comment that will certainly be useful to readers. The detailed information of this module and solar cells can be found in the newly-added references, but I also added the essence of the device configuration, material (including bandgap), and process.

Very general consideration of the radiative coupling does not provide any insight on this problem from the material related point of view: it is not clear which materials/structures can provide most effective degree of the radiative coupling. Some concrete experimental facts, apart general statements like (typically 15%, line 245) should be provided.

I appreciate your comment. It is true this evidence is crucial for validation of the model. The evidence, including reference, was added.

The material issue that realizes good ERE is also crucial to the reader’s interests. I added related content from the introduction sections with several new references and material target indicators (defect density).

Reviewer 2 Report

1. What does ELO stand for in line 24?

2. The descriptions of many sentences throughout entire article are ambiguous so that the readers may not be able to catch the exact meanings of these descriptions. Some examples are given as follows. Please revise properly.

Line 38-39, However, the materials and process in the real world were not ideal, and the record efficiency values of photovoltaic are less than that [5-6].

Line 102, The radiative recombination was also identified to the performance of the multi-junction cell, even in operation under the standard testing condition, thus a single pattern of the spectrum.

Line 108, Moreover, even emerging solar cells like Perovskite solar cell, the radiative coupling and photon recycle was identified as it could not be ignored [69].

3. Few assumptions for the model are described from line 190 to line 194. Are there any other assumptions?

4. What are the parameters and assumptions for the non-concentrating super-multi-junction solar cell reaching the maximum potential efficiency (50 %) under the realistic conditions? If any, a summary table for these parameters and assumptions is suggested to present.

5. Are there any practical methods and/or experimental results demonstrating that the radiative efficiency can reach nearly 100%? What are the radiative efficiencies for the current experimental results?

6. Line 412, “one is the optical cap layer that may be the directional photon coupling layer in the application of the photonics technologies”

What are the functions of the optical cap layer for the super-multi-junction solar cells? Are there any references describing and applying this technology?

7. For the performance ratios presented in Fig. 4 and Fig. 5, what is the definition of the performance ratio?

8. Line 144, “The annual energy yield of the multi-junction cells is not always boosted by the number of junctions.”

Are there any references or experimental results supporting this argument?

Author Response

Dear Reviewer 2,

I appreciate your constructive suggestions. Due to the constrains of the browser, images were not pasted in this documents. I am wondering if you could find an attached file for my complete reply. 

What does ELO stand for in line 24?

I appreciate your suggestion for the correction of the flaw. Now, it is corrected.

The descriptions of many sentences throughout entire article are ambiguous so that the readers may not be able to catch the exact meanings of these descriptions. Some examples are given as follows. Please revise properly.

I appreciate your suggestion. The entire texts were carefully examined, and we corrected many flaws. Please, see our correction report (another file).

Line 38-39, However, the materials and process in the real world were not ideal, and the record efficiency values of photovoltaic are less than that [5-6].

I appreciate your suggestion. The corresponded correction is as follows.

Line 102, The radiative recombination was also identified to the performance of the multi-junction cell, even in operation under the standard testing condition, thus a single pattern of the spectrum.

I appreciate your suggestion. The corresponded correction is as follows.

Line 108, Moreover, even emerging solar cells like Perovskite solar cell, the radiative coupling and photon recycle was identified as it could not be ignored [69].

I appreciate your suggestion. The corresponded correction is as follows.

Few assumptions for the model are described from line 190 to line 194. Are there any other assumptions?

I appreciate your suggestion. It is also a crucial technological information to most of the readers. I added some texts and two figures.

What are the parameters and assumptions for the non-concentrating super-multi-junction solar cell reaching the maximum potential efficiency (50 %) under the realistic conditions? If any, a summary table for these parameters and assumptions is suggested to present.

I appreciate your suggestion. It is also a crucial technological information to most of the readers. I added some texts and a table.

Are there any practical methods and/or experimental results demonstrating that the radiative efficiency can reach nearly 100%? What are the radiative efficiencies for the current experimental results?

I appreciate your comment. In the measurement by laser, the increased ERE was observed. The highest ERE in the solar cell I have ever found in the literature (reviewed paper) is 35 %.

The current commercial level of solar cell technology is not as high as that value. The level of the current technology was found in several papers, and I added them in the reference list.

Line 412, “one is the optical cap layer that may be the directional photon coupling layer in the application of the photonics technologies”

What are the functions of the optical cap layer for the super-multi-junction solar cells? Are there any references describing and applying this technology?

I appreciate your suggestion, useful to readers to understand the technology contents. Several texts, not only line 412, were added.

For the performance ratios presented in Fig. 4 and Fig. 5, what is the definition of the performance ratio?

I appreciate your excellent comment. Since it was monitored in the DC system, the definition of the performance ratio is slightly different from the IEC61724 that is exclusively applied to the AC system. The definition of the performance ratio was added to each caption of the figure.

Line 144, “The annual energy yield of the multi-junction cells is not always boosted by the number of junctions.”

Are there any references or experimental results supporting this argument?

I appreciate your comment. One reference was added that claimed that the efficiency of the multi-junction might drop by the increase of the number of junctions.

Round 2

Reviewer 1 Report

Authors have provided answers, have made relevant corrections and have addressed raised up questions and comments.  

An improved version of the article can be published now.

Author Response

We appreciate your constructive review.

Reviewer 2 Report

1. The authors have cited the references about the values of ERE for some solar cells. To the authors’ best knowledge, it was reported that the greatest value of ERE is 35%. Obviously, ERE of around 100% is not realistic. Please indicate the reported value of 35% for ERE in the article.   

2. Line 448-450, and Line 455-458, there are redundancies.

3. Line 162-162, the authors stated that the combination of the numerical optimization calculation and Monte Carol method was employed for the performance analysis. Although some references have been cited therein, please describe how to combine the numerical optimization calculation and the Monte Carlo method for the design, performance analysis, and optimization calculation.

Author Response

We appreciate the constructive suggestions. Besides specific comments, we improved the English and Style (Fonts). Please, see the attached file for the correction record (highlighted).

The authors have cited the references about the values of ERE for some solar cells. To the authors’ best knowledge, it was reported that the greatest value of ERE is 35%. Obviously, ERE of around 100% is not realistic. Please indicate the reported value of 35% for ERE in the article.

We appreciate the constructive suggestions. We added following texts. Please, see the attached file for the correction record (highlighted).

The best measured ERE in the best knowledge of authors is 35 % [108]. Obviously it is far less than 100 %.

Line 448-450, and Line 455-458, there are redundancies.

We appreciate the constructive suggestions. We removed the redundant texts in line 448 – 450. Please, see the attached file for the correction record (highlighted).

Line 162-162, the authors stated that the combination of the numerical optimization calculation and Monte Carol method was employed for the performance analysis. Although some references have been cited therein, please describe how to combine the numerical optimization calculation and the Monte Carlo method for the design, performance analysis, and optimization calculation.

We appreciate the constructive suggestions. We added following texts. Please, see the attached file for the correction record (highlighted).

The vector of variables in numerical optimization is the set of bandgap energy of each junction. In each moment (time and date), the meteorological parameters and atmospheric parameters were given by random numbers (Monte Carlo method). The energy output was summed in a year and divided by the total irradiation, also calculated by the above meteorological and atmospheric conditions.
